# The Frequency of a Magnetic Field Determines the Behavior of Tumor and Non-Tumor Nerve Cell Models

**DOI:** 10.3390/ijms26052032

**Published:** 2025-02-26

**Authors:** Isabel López de Mingo, Marco-Xavier Rivera González, Milagros Ramos Gómez, Ceferino Maestú Unturbe

**Affiliations:** 1Escuela Técnica Superior de Ingenieros de Telecomunicación (ETSIT), Universidad Politécnica de Madrid, 28040 Madrid, Spain; isabel.lopez@ctb.upm.es (I.L.d.M.); milagros.ramos@ctb.upm.es (M.R.G.); 2Centro de Tecnología Biomédica (CTB), Universidad Politécnica de Madrid, 28223 Madrid, Spain; marco.rivera@ctb.upm.es; 3Escuela Técnica Superior de Ingenieros Informáticos (ETSIINF), Universidad Politécnica de Madrid, 28223 Madrid, Spain; 4Centro de Investigación Biomédica en Red (CIBER-BBN), 28029 Madrid, Spain

**Keywords:** magnetic field, glioblastoma, neuroblastoma, astrocytes, biological window, viability, proliferation, frequency

## Abstract

The involvement of magnetic fields in basic cellular processes has been studied for years. Most studies focus their results on a single frequency and intensity. Intensity has long been the central parameter in hypotheses of interaction between cells and magnetic fields; however, frequency has always played a secondary role. The main objective of this study was to obtain a specific frequency that allows a reduction in the viability and proliferation of glioblastoma (CT2A) and neuroblastoma (N2A) cell models. These were compared with an astrocyte cell model (C8D1A) (nontumor) to determine whether there is a specific frequency of response for each of the cell lines used. The CT2A, C8D1A, and N2A cell lines were exposed to a magnetic field of 100 µT and a variable frequency range between 20 and 100 Hz for 24, 48 and 72 h. The results fit a biological window model in which the viability and proliferation of N2A and CT2A cells decrease statistically significantly in a 50 Hz center of value window. In addition, the non-tumor cell model showed different behavior from tumor cell models depending on the applied frequency. These results are promising in the use of magnetic fields for therapeutic purposes.

## 1. Introduction

Cells of the nervous system have been one of the most widely used models in magnetic field exposure assays since in vitro and in vivo assays in magnetobiology began to be published. One of the main interests of the use of these cell lines in this discipline is their excitable character. Astrocytes are the most numerous and principal cells in the set of glial cells of the nervous system [1,2,3,4]. They have long been considered as support cells for neurons with no other function that gave them greater importance than that of maintaining the main cell of nerve communication [4]. Recent years have seen a growing interest in the study of astrocyte function due to biotechnological advances that have led to an understanding of their heterogeneity and their relevance in health and pathology [5,6,7,8,9]. Astrocytes play an important role in key processes such as homeostasis (neurotransmitter recycling, ionic balance, modulation of synaptogenesis, synaptic transmission, maintenance of the blood–brain barrier) and contribute greatly to central nervous system (CNS) repair [10,11,12]. The importance of astrocytes has been described in numerous neurodegenerative pathologies such as multiple sclerosis [13,14,15,16,17,18,19,20,21,22,23], Alzheimer’s disease [24,25,26], Parkinson’s disease [27,28] and Huntington’s disease [29,30,31,32], but also in tumors such as gliomas [33,34,35,36]. Malignant astrocytic gliomas are the most common intracranial tumors [35,36]. In particular, glioblastoma (grade IV astrocytoma according to the classification of the World Health Organization) is the most lethal glioma and currently has no cure [37,38].

The relationship between magnetic fields and cancer has been studied for years [39,40,41,42,43,44,45,46,47,48]. In vitro studies of magnetic field exposure have explored various cellular processes such as viability, proliferation or apoptosis, which are closely related to tumor progression [49,50,51,52,53,54,55,56,57]. There is a dual objective in in vitro experiments using tumor cells, on the one hand, to reduce tumor cell proliferation and viability to consider magnetic fields as a therapeutic alternative, and, on the other hand, to determine the possible adverse effects on tumor progression that prolonged exposure to extremely low electromagnetic fields (ELF-EMFs) could cause [41,47,48,58,59,60]. However, most of the results are inconclusive or incomplete. This, together with the lack of knowledge of the mechanisms of action underlying the occurrence of cellular effects, means that in vitro experimentation requires further exploration. One of the main errors highlighted by many experts in the field is directly related to the methodologies and choice of exposure parameters (frequency, intensity, exposure time, waveform) [61,62,63,64]. Cells have been shown to have different behaviors depending on the value of the chosen exposure parameter and that careful exploration of each of these parameters should be performed [65]. The chosen exposure parameters can determine the cell response.

One of the most important exposure parameters is frequency [65]. It has been suggested for years that frequency plays a key role in producing different responses in cell lines and that different cell types may respond to specific frequencies and is a key element in the development of many interaction mechanisms such as theories based on resonance effects [66,67,68,69,70,71,72,73,74,75]. Although the determining parameter in the development of exposure regulations and interaction models has been intensity, with cellular energy deposition as the key factor for the occurrence of effects, frequency has been and is one of the parameters that has attracted the most interest in many research teams [65,76].

The main objective of this article is to obtain a specific frequency of a 100 µT intensity field that will reduce the viability and proliferation of glioblastoma and neuroblastoma cells. The results of this cell line will be compared with astrocytes (nontumor cells) in order to determine whether there is a specific frequency of response for each of the cell lines used and, therefore, whether the cells are sensitive to a specific combination of exposure parameters.

## 2. Results

### 2.1. Viability

#### 2.1.1. Assay 1: Search for a Frequency Window in the Range (0–100) Hz

For the determination of a frequency window in the frequency range 0–100 Hz, the MTT metabolic activity assay is performed on three cell lines whose origin is from nervous tissue of different nature, tumor (CT2A, N2A) and non-tumor (C8D1A). The results of the percentage of viability by frequencies and exposure times can be seen in Figure 1A,D,G.

The results obtained in the second reduced window of exposure frequencies showed the most significant decrease in tumor lines at frequencies close to 50 Hz, so the window reduction was carried out around this frequency.

At 20 Hz, the results show a significant decrease in viability for both tumor lines in all three exposure time bands. For CT2A cells, the greatest decrease in percent viability occurs at 72 h (−25.91 ± 3.78%, *p* < 0.001), with the greatest decrease for N2A at 48 h (−20.41 ± 2.51%; *p* < 0.001). The C8D1A cell line only shows statistically significant results of increased percentage viability in 24 h (12.06 ± 3.12%; *p* < 0.001).

For a frequency of 40 Hz, CT2A cells show a close decrease in viability in the three time intervals studied, which is lower at 24 h of exposure (−16.87 ± 0.90%; *p* < 0.001). In the case of neuroblastoma cells, the results found at 20 Hz are maintained, with the greatest decrease at 24 h (−18.59 ± 1.84%; *p* < 0.001). Astrocytes, on the other hand, further increase their viability (20.51 ± 4.98%; *p* < 0.001) with respect to the results obtained 24 h after exposure to 20 Hz. Exposures at 48 h (12.31 ± 2.27%; *p* = 0.079) and 72 h (4.32 ± 6.58%; *p* = 0.105) do not obtain statistically significant results for the non-tumor line.

At 60 Hz, tumor cells again show a decrease in percentage viability which is repeated, this time in non-tumor cells. CT2A cells show a greater decrease in viability at 24 h (−24.21 ± 1.51%; *p* < 0.001) and 72 h (−24.29 ± 3.68%; *p* < 0.001), while N2A cells show it at 24 h of exposure (−25.50 ± 2.68%; *p* < 0.001). In the case of astrocytes, they show a decrease in viability for all three exposure time intervals, with minimum viability at 24 h (−25.50 ± 2.68%; *p* < 0.001).

At 80 Hz exposures, the tumor cells tested begin to behave differently. For CT2A cells, exposures of 24 h (−19.57 ± 7.04%; *p* < 0.001) and 72 h (−17.12 ± 2.88%; *p* < 0.001) produce statistically significant decreases in viability, but this does not occur at 48 h (−1.95 ± 3.72%; *p* = 0.182). N2A cells, on the other hand, only show statistically significant results of increased viability 72 h after exposure (13.16 ± 10.66%; *p* = 0.01). Astrocytes show a viability reduction that is more moderate than that found at 60 Hz, being very similar at all three exposure times, with the greatest decrease at 24 h of exposure (−11.29 ± 2.07%; *p* < 0.001).

Finally, exposure to a frequency of 100 Hz produces in CT2A cells a statistically significant decrease after 72 h of exposure (−18.50 ± 2.64%; *p* < 0.001), but not statistically significant results at 24 h (0.50 ± 1.92%; *p* = 0.105) nor at 48 h (−1.65 ± 2.98%; *p* = 0.141). In the case of astrocytes, the reduction in viability was maintained, with the lowest percentage found at 24 h (−12.79 ± 2.47%; *p* < 0.001) and 48 h (−13.15 ± 3.68%; *p* < 0.001).

Considering the results obtained from the first frequency screening, we decided to incorporate the frequencies of 30 and 50 Hz, in order to explore in greater depth what happens at the intermediate points of the frequency window formed by 20, 40 and 60 Hz, which, in the case of glioblastoma and neuroblastoma cells, managed to reduce their viability at all the exposure times tested. For this purpose, the frequency window scan was reduced to a range of (20–60) Hz by dividing the frequency intervals by 10 Hz.

#### 2.1.2. Assay 2: Search for a Frequency Window in the Range (20–60) Hz

Following the results obtained in the first frequency window (Figure 1A,D,G), it was decided to reduce the window to a frequency range of (20–60) Hz by dividing the frequency steps by 10 Hz. Therefore, the frequencies 30 and 50 Hz were incorporated (Figure 1B,E,H).

The results obtained at 30 Hz exposures show in glioblastoma a considerable reduction in viability with respect to the frequencies preceding it, 20 Hz, and after it, 40 Hz. The minimum percentage of viability is shown for exposure times of 24 h (−35.61 ± 3.93%; *p* < 0.001). In the case of neuroblastoma cells, 30 Hz shows very similar results in reducing the percentage of viability with respect to frequencies of 20 and 40 Hz. The three exposure times converge to similar viability percentage values (24 h: −18.42 ± 3.01%; 48 h: −16.00 ± 5.35%; 72 h: −17.54 ± 6.09%; *p* < 0.001). For non-tumor cells, C8D1A, the 30 Hz frequency produces a considerable decrease in viability compared to the 20 and 40 Hz frequencies, with the minimum percentage at 72 h (−16.12 ± 5.05%; *p* < 0.001).

At 50 Hz exposures, glioblastoma cells show the greatest decrease in the percentage of viability shown so far, converging the three hours of exposure to near viability values (24 h: −35.61 ± 3.93%; 48 h: −33.95 ± 5.20%; 72 h: −32.87 ± 1.85%; *p* < 0.001). In the case of neuroblastoma, it is also at 50 Hz, the frequency at which a greater reduction in viability is observed, being the lowest at 48 h (−14.02 ± 4.05%; *p* < 0.001). Astrocytes also show a decrease in the percentage of viability, less pronounced than at 60 Hz, with a minimum at 72 h of exposure (−18.04 ± 5.33%; *p* < 0.001).

#### 2.1.3. Assay 3: Search for a Frequency Window in the Range (40–60) Hz

Following the results obtained in the second frequency window (Figure 1B,E,H), the frequency window is again reduced to the range (40–60) Hz in 5 Hz steps, so that the frequencies of 45 and 55 Hz are incorporated into the test. The viability results obtained can be found in Figure 1C,F,I.

At 45 Hz, glioblastoma cells reduce their viability considerably at 72 h (−41.14 ± 2.69%; *p* < 0.001), although in the rest of the hours tested, it is maintained with percentages similar to those obtained at 50 Hz (24 h: −28.47 ± 11.94%; 48 h: −29.47 ± 6.04%; *p* < 0.001). In the case of neuroblastoma lines, a viability reduction similar to that found at 40 Hz is shown (24 h: −9.99 ± 4.69%; 48 h: −14.02 ± 4.05%; 72 h: −17.00 ± 7.92%; *p* < 0.001). Astrocytes show a decrease in viability at this frequency with the lowest percentage at 72 h (−24.31 ± 3.92%; *p* < 0.001).

In the case of exposures of 55 Hz, CT2A cells reduce the percentage of viability to the minimum at 72 h (−38.32 ± 1.93%; *p* < 0.001). In the case of N2A cells, they maintain a viability percentage close to that found at 45 and 50 Hz with the minimum at 48 h (−25.13 ± 3.38%; *p* < 0.001), but with very similar viability percentages at all three hours tested. Non-tumor cells show the greatest decrease in the percentage of viability for this frequency, with very similar values for the three hours of exposure, with the lowest percentage at 48 h (−28.14 ± 2.64%; *p* < 0.001) and 72 h (−27.79 ± 1.58%; *p* < 0.001).

### 2.2. Proliferation

Proliferation assays are performed in the frequency window in which the most statistically significant results of viability reduction have been obtained, i.e., the frequencies of 45, 50 and 55 Hz are chosen. The results can be seen in Figure 2.

Glioblastoma cells (CT2A) show a decrease in proliferation results at all three frequencies tested, decreasing with increasing frequency, i.e., at 45 Hz (−16.06 ± 1.15%; *p* < 0.001), 50 Hz (−19.43 ± 2.68%; *p* < 0.001) and 55 Hz (−26.24 ± 2.20%; *p* < 0.001) (Figure 2A). In neuroblastoma cells (N2A), the results show a decrease in the percentage of proliferation at the three frequencies, which, unlike glioblastoma cells, decreases with frequency, i.e., at 45 Hz (−25.99 ± 1.51%; *p* < 0.001), 50 Hz (−26.29 ± 0.68%; *p* < 0.001) and 55 Hz (−21.64 ± 0.48%; *p* < 0.001) (Figure 2A). For non-tumor cells, astrocytes, the response at 45 Hz shows an increase in proliferation of 11.80 ± 2.48% (*p* = 0.003), which results in a reduction in proliferation at 50 Hz (−24.87 ± 5.94%; *p* < 0.001) and 55 Hz (−21.52 ± 2.09%; *p* < 0.001) (Figure 2A).

In the same assay, the number of dead cells contained in the different samples is determined (Figure 2B). Glioblastoma cells (CT2A) show a single example of significant data on the reduction in dead cells at 45 Hz with respect to the control (−15.12 ± 1.41%; *p* = 0.04). The rest of the frequencies show no significant data, i.e., at 50 Hz (1.39 ± 3.49%; *p* = 0.839) and 55 Hz (5.21 ± 2.98%; *p* = 0.542). Neuroblastoma cells do not show significant results for any of the frequencies studied. Astrocytes show an increase in dead cells at 45 Hz (54.76 ± 2.26%; *p* < 0.001), which decreases the significance of the increase in dead cells at 50 Hz (15.11 ± 0.66%; *p* < 0.001) and at 55 Hz (2.57 ± 2.56%; *p* = 0.01).

### 2.3. Apoptosis

The fluorescence results show a statistically significant decrease in cell viability at the 50 Hz frequency for the three tumor cell models exposed to the magnetic field (CT2A: *p* < 0.001; C8D1A: *p* = 0.027; N2A: *p* = 0.001) (Figure 3M). The rest of the frequencies do not obtain statistically significant results, except for the increase for N2A at the frequency of 55 Hz (*p* = 0.002) (Figure 3L). In the case of proliferation, a significant decrease is shown at 50 Hz for the three cell models (CT2A: *p* < 0.001; C8D1A: *p* < 0.001; N2A: *p* < 0.001) (Figure 3N). In the astrocyte cell model, a statistically significant increase (*p* = 0.004) in proliferation at 45 Hz is shown (Figure 3F). In the neuroblastoma cell model, a statistically significant decrease is shown at 55 Hz (*p* < 0.001) (Figure 3L). Regarding the number of dead cells, only the neuroblastoma model obtains significant results of a decrease the in dead cells at 55 Hz (*p* < 0.001) (Figure 3O).

## 3. Discussion

The main objective of this research was to find a target frequency that reduces the proliferation and viability of two tumor cell models of nervous tissue, CT2A (glioblastoma) and N2A (neuroblastoma) and to determine if the cell behavior against the same frequency is similar using a non-tumor cell model of the same tissue, in this case glia cells and astrocytes (C8D1A). The results show a dependence of the percentages of viability, proliferation and apoptosis with respect to the selected frequency that conforms to a “biological window” model in which the results appear in determined ranges of the variant exposure parameter and outside these ranges, the effect on the cellular process disappears or diminishes considerably. A value of 50 Hz has been determined as the target frequency that produces the greatest decrease in viability for the glioblastoma cell model (CT2A) when looking at the three exposure times tested (24, 48 and 72 h). Applying the 50 Hz frequency reduces the viability of CT2A tumor cells by more than 30% and their proliferation by more than 20%. This phenomenon is repeated with similar percentages in the neuroblastoma cell model (N2A). In comparison with nontumor cells, C8D1A shows a decrease in viability also at 50 Hz; however, the reduction is smaller and their response for lower frequencies, 20 and 40 Hz, is an increase in viability, but a reduction for the CT2A and N2A tumor models. One of the most widely supported explanatory theories of magnetobiology over the years has been the existence of biological windows, which are specific resonance conditions that produce alterations in cellular processes when a magnetic field with a certain frequency and intensity is applied to cellular systems over a very specific range of these parameters for a certain time [76,77,78,79,80,81,82,83,84]. This concept was introduced by Ross Adey in 1975, but these effects had been observed in earlier experiments [85,86,87]. The existence of frequency, intensity and time windows that could be grouped under the term “biological window” has been described [76,78,88]. The biological window is a means by which the living organism detects electromagnetic fields, but it does not only refer to a single frequency, intensity and time of exposure, but to a specific combination of these parameters, a code, which after exposure returns a certain cellular response [77]. The cell could be understood as a sensor capable of recognizing and being sensitive to certain magnetic field parameters [73,82]. The results presented fit a frequency window model in which certain frequency values produce effects on viability and proliferation processes specific to each of the cell lines used.

Despite being a main parameter in the development of interaction models and mechanisms of action [66,67,68,69,70,71,72,73,74,75,89], at the experimental level, there are few studies that use different values of frequency in order to obtain the characterization of cell behavior based on this parameter or that justify the value chosen to carry out their experiments [65]. Most of these studies use frequencies of 50 Hz [60,90,91,92,93,94,95,96,97,98,99,100,101,102,103,104,105,106,107,108,109,110,111,112,113,114,115] and 60 Hz [60,116,117,118,119,120,121,122,123], mainly because these are used in the electrical distribution network across the world. Bergandi et al. (2022) exposed cells from a pancreatic tumor model (PANC-1), glioblastoma primaries and a breast tumor model (MCF-7) to a 70 µT magnetic field with varying frequencies of 3, 4, 6, 10 and 14 Hz, finding that the proliferation results were dependent on the frequency applied [124]. García-Minguillán et al. (2019) exposed a murine glioblastoma cell model (CT2A) to a magnetic field of 30 µT and frequency of 7.8, 14, 20, 26, 33, 39, 45 and 51 Hz and observed that viability behavior was dependent on the applied frequency [125]. In another experimental setup used in the study, they applied a magnetic field of 100 µT with a variable frequency of 20, 30 and 50 Hz that showed a significant decrease in viability for the 30 Hz frequency, but not for the rest, confirming the dependence of the cellular results on the selected frequency.

For decades, due to the establishment of exposure regulations for ELF-EMFs, intensity has been believed to be the determining parameter for most of the effects found in both in vitro and in vivo experiments [126,127,128,129]. The so-called “dose effect”, in which the probability of effects increases proportionally to intensity, has been one of the most reinforced theories in the scientific literature [126,127,128,129]. However, biological systems are not linear systems [130,131,132], which leads us to believe that even if the stimulus increases, the response will not. The results presented show that when the same intensity is used, different results are obtained depending on the frequency applied. Thus, frequency and intensity are deduced to have a strong dependence on the cellular response and that the increase in the occurrence of the results is far from being dependent only on the applied intensity. This had already been observed in other previously published studies. Nazamtaheri, M. et al. (2022) exposed cells from a model of human prostate carcinoma (DU145), human breast carcinoma (MDA-MB-231), human umbilical cord endothelial (HUVEC) and chronic leukemia (K562) to a magnetic field of 1, 10 and 100 mT and varying frequency of 0. 01, 0.1, 1 and 10 Hz, concluding that significant apoptosis and viability results were found for a specific combination of frequency and intensity that was dependent on each cell line used [133].

For many years, attempts have been made to uncover the mechanism underlying the occurrence of biological effects in the presence of a magnetic field [73,74,75,89,134,135,136,137,138,139,140,141]. Despite efforts to develop physico-mathematical models, no model has been found that explains each of the effects found in cell experimentation. Researchers agree that the cell membrane is the first place where ELF–EMF interaction occurs [77,82,83,142,143]. The cell membrane is an electrical insulator ($\varepsilon_{r}\approx 6$) that does not allow the electrical component of the ELF–EMF to pass through, so it seems evident that the main component that is altering cell behavior is the magnetic one [132,144]. The cell membrane has dual functionality, acting as a sensor and also as an effector in the presence of a magnetic field [74,82,132]. As a sensor, it detects chemical alterations in intra- and extracellular fluid and mediates cell signaling. As an effector, it transmits electrical and chemical signals to neighboring cells co-communicating with diverse cellular ensembles. Communication phenomena are basically membrane phenomena that are produced by transduction complexes (specific receptor proteins, transducer proteins, enzymes) that amplify the weak signal that is promoted by the binding of signaling molecules to their specific receptors on the extracellular side of the cell membrane [82]. Consequently, effector molecules are generated at the cytosolic level (second messenger) that induce various metabolic changes [132]. Therefore, once the first interaction with the membrane has occurred, it is likely that these secondary mechanisms produce the observed effects through the alteration of signaling pathways. The interaction of magnetic fields and cell membrane proteins appears to result primarily from electronic polarization, reorientation of dipole groups, and changes in the concentrations of charged species in the vicinity of charges and dipoles [132]. It has been proposed that these parameter codes acting on living systems could be influencing cell signaling cascades, with the cell membrane being one of the main targets of interaction, in processes such as the transport of neurotransmitters, hormones, expression of growth regulatory enzymes and cancer-promoting chemicals [77,82,83,142,143]. This coding would occur mainly in the frequency/intensity combination of the applied signal. Cell proliferation is a highly dynamic process regulated by different signaling pathways, such as PI3K/Akt, MAPK/ERK, Wnt/ß-catetenin, JAK/STAT, calcium/calmodulin, and with the participation of numerous down-regulating molecules and checkpoints such as the p53 pathway (inhibits proliferation in response to DNA damage), pRB/E2F (controls cell cycle progression) or the cyclin inhibitors p21 and p27 (regulate cell cycle progression) [145]. The alteration of cell proliferation because of magnetic field exposure can be altered both by the initiation of apoptosis mechanisms, programmed cell death, and by the interruption of proliferation as a consequence of maintaining the cell in an inactive or senescent state [146]. Magnetic fields have been shown to alter the different signaling pathways mentioned above. Xu, A. et al. (2020) exposed different breast cancer cell models to a magnetic field of 1 mT and frequencies of 50, 125, 200 and 275 Hz, and determined that proliferation decreased as a function of exposure time and an optimal frequency of 200 Hz [147]. Their results showed suppression of the PI3K/AKT cell signaling pathway, through the increase in ROS levels within the cell as a consequence of exposure to the magnetic field used. Also, Patruno, A. et al. (2015) showed that in a keratinocyte cell model exposed to 50 Hz and 1 mT, the response of the PI3K/Akt signaling pathway was modulated through the mTOR molecule and the activation of the MAPK/ERK pathway, another pathway responsible for the regulation of cell proliferation, was promoted [148]. Lange S. et al. (2001) exposed a human cell model of amniotic cells to a magnetic field of 50 Hz and 1 mT and found that there was a decrease in cyclin D1 expression and an increase in p21 expression, so that the G_1_-phase of the cell cycle was inhibited as a consequence of the exposure [149]. The alteration of p21 expression was also confirmed by other authors [150,151]. Mehdizadeh, R. et al. (2023) exposed a model of glioblastoma cells to a magnetic field of 100 mT and 1 Hz, observing increased expression of p53 and p21 inducing cycle arrest in G_2_/M-phase and thus initiating apoptosis processes [152].

One of the hypotheses that could explain the effects found is the existence of a bioactive code, understood as a specific combination of magnetic field exposure parameters (frequency, intensity, and exposure time), which the cell recognizes, and which would represent the cusp of a window response in a specific range of these exposure parameters. Thus, the first site of interaction with the cell would be the cell membrane, where the signal would be amplified. This signal would produce changes in signaling cascades that would result in the alteration of measurable biological processes such as viability and proliferation. Therefore, this code, this specific combination of exposure parameters, could be understood as a resonant condition that would produce alterations in cell behavior. The existence of a frequency–intensity–time code that can modify the cellular behavior of cells from nervous tissue in the processes of viability and proliferation allows the use of magnetic fields in therapeutic applications in pathologies such as neurodegenerative diseases and tumor processes. Some authors state the so-called “window of opportunity” or “therapeutic window” as that combination of exposure parameters that could alter a pathological process through a therapeutic action [81]. In addition, many point out that exposure to magnetic fields in an out-of-equilibrium system, in a pathological state, could be more effective than those applied to systems in equilibrium or healthy controls [81]. According to the results presented, it is also observed that nervous tissue cells show characteristic behaviors in response to the magnetic field and that it has been possible to reduce the proliferation and viability of the glioblastoma and neuroblastoma cell model by applying a magnetic field with a frequency of 50 Hz and 100 µT of intensity. Furthermore, the behavior of astrocytes has been modulated. Astrocytes are involved in numerous vital cellular processes, such as homeostasis (neurotransmitter recycling, ionic balance, modulation of synaptogenesis, synaptic transmission, maintenance of the blood–brain barrier), contributing greatly to CNS repair [10,11,12]. The importance of astrocytes has been described in numerous neurodegenerative pathologies such as multiple sclerosis [13,14,15,16,17,18,19,20,21,22,23], Alzheimer’s disease [24,25,26], Parkinson’s disease [27,28] and Huntington’s disease [29,30,31,32], but also in tumors such as gliomas [33,34,35,36]. In pathologies such as Alzheimer’s disease and Parkinson’s disease, astrocytic networks become dysfunctional due to oxidative stress, chronic inflammation, or accumulation of toxic proteins resulting in impaired synaptic and metabolic hemostasis. According to the results presented, frequency could succeed in increasing the viability and proliferation of astrocytes to prevent progressive elimination of networks in neurodegenerative diseases and reduce tumor cell proliferation.

## 4. Materials and Methods

### 4.1. Cell Cultures

Mouse cell lines CT2A (glioblastoma), N2A (neuroblastoma) and C8D1A (astrocytes) are used. The CT2A and N2A lines are on loan from the Cajal Institute of Madrid belonging to the Centro Superior de Investigaciones Científicas. The C8D1A line (ATCC number: CRL-2541) was obtained from the ATCC (American Type Culture Collection, LGC Standards, Teddington, UK). The three cell lines were grown in monolayer culture in Dulbecco’s modified Eagle’s medium with elevated glucose (DMEM) (DDBiolab, w/L-Glutamine, no sodium pyruvate, cat. no. L0102-500, Barcelona, Spain), supplemented with 10% fetal bovine serum (DDBiolab, cat. no. P30-3302, Barcelona, Spain), 1% L-Glutamine (DDBiolab, 200 mM, cat. no. P04-80100) and 1% penicillin/streptomycin (DDBiolab, penicillin 5000 Ul/mL, streptomycin 5). All cell lines were cultured at 37 °C under an atmosphere of 5% CO_2_ in air. Cell subpopulations were prepared through the cell passaging technique 2 times a week when the culture plates were close to reaching 90% confluence.

### 4.2. Electromagnetic Field Exposure System

The RILZ coil system, designed specifically for the commissioning of cellular studies, is used (Figure 4) [153]. In summary, it consists of two coils, each in the shape of a capsule consisting of two semicircles with a radius of 5 cm joined by two 10 cm long straight lines. The distance between the two coils is 3.5 cm and the width for winding is 7 cm. They are covered by 222 turns of 18 AWG 18 gauge enameled copper. In the center of the coils, there is a support for the cell plates, which ensures that the cell plates are always in the center of the configuration to guarantee field homogeneity. The electronic system that powers the coils is also of our own design and specific to the cellular application. It generates a square signal controlled by a microprocessor. The frequency and current values can be displayed on an LCD screen and set by precision potentiometers connected to the analog-to-digital converter (ADC) of the microcontroller. The frequency is set between DC and 200 Hz, using interrupts generated by the microcontroller’s internal timer. The current intensity is set between 0 and 2 Amps, using a current source controlled by the microcontroller and by means of a power mosfet fed to the RILZ coils. The magnetic field is monitored using a Model 480 Fluxmeter from LakeShore (Lake Shore Cryotronics©, Westerville, OH, USA) with a Model MMZ-2502-UH triaxial probe (Lake Shore, Cryotronics©, Westerville, OH, USA).

The RILZ exposure system was introduced into a Thermo Scientific 3111 series II incubator (Thermo Fisher Scientific Inc., Waltham, MA, USA) at the time of cell assays to maintain cells under physiological conditions of 37 °C temperature and 5% CO_2_. Electromagnetic field intensity measurements were performed with stimulation equipment in the incubator using the LakeShore Model 480 Fluxmeter (Lake Shore Cryotronics©, Westerville, OH, USA) and the triaxial probe model MMZ-2502-UH (Lake Shore Cryotronics©, Westerville, OH, USA) with the absolute intensity value recorded in the three directions of x (23.14 ± 0.39 µT), y (42.30 ± 1.19 µT) and z (5.96 ± 0.49 µT) space. The coils were raised 5 cm on non-metallic supports with respect to the metal tray of the incubator, so they were not in direct contact with it in order to eliminate the noise generated by the induction of the magnetic field by direct contact with the grounded metal surface.

### 4.3. Exposure Conditions

The stimulation magnetic field had a fixed intensity of 100 µT (exposure limit established by recommendation 1999/519/EC in Spain for a frequency of 50 Hz) in all tests performed [154]. Frequency and exposure time were variable parameters (Table 1). It was decided to use 24, 48 and 72 h of exposure because these are the time intervals in which most in vitro experiments are grouped according to a previously published review [65].

Different cell tests were performed on the basis of the results obtained in the tests preceding them. The different configurations of the exposure parameters and the assay performed can be seen in Table 1. In all configurations, a train of square pulses was used as a waveform and an uninterrupted exposure. Cells were seeded in the greater than 97% homogeneity area delivered by the RILZ system and given by a previous publication [153].

For each of the cell assays performed and presented in Table 1, the corresponding controls were performed. The control group was under the same experimental conditions (temperature, humidity, relative position in the incubator), but the exposure system was kept off, that is, the coils were not fed. The same number of control replicas was performed as in the evaluation of exposure conditions.

### 4.4. Metabolic Activity Test

To perform the metabolic activity assay, cells were seeded in 96-well plates at a final concentration of 60,000 cells/mL. The 3-(4,5-dimethylthiazol-2-yl)-2,5-diphenyl-2H-tetrazolium bromide (MTT) assay (Biotium, Fremont, CA, USA, MTT Cell Viability Assay Kit, cat. no. 30006) was used. The assay was performed according to the manufacturer’s instructions. In summary, 10 µL of MTT agent was added to each well of the 96-well plate. Next, the plates were incubated in the dark for 4 h (temperature of 37 °C). After this period, 200 µL of dimethyl sulfoxide (DMSO, Corning Media Tech, New York, NY, USA, Cat. no. 15303671) was added to each of the wells and resuspended to break up the crystals formed. The concentration of seeded cells was 60,000 cells/mL, and they were exposed 24 h after seeding.

The absorbance was measured using a HEALES microplate reader model MB-580 (HEALES, Shenzhen, China) at the wavelengths of 570 nm and 630 nm (corresponding to the background signal). The graphical representation of the absorbance results obtained is performed with Prism 9 software (GraphPad Software, v.9.3.1, Boston, MA, USA). A total of 10 replicates were performed in two independent experiments for each of the frequencies, including the controls. For data analysis, the results of the subtraction of absorbances were ordered from highest to lowest at each of the frequencies (including the no exposure controls) and the percentage of viability was calculated as follows:$${Viability\left(\%\right)}_{i}=\frac{{Abs}_{570-630nm}({Exp}_{i})}{{Abs}_{570-630nm}({Control}_{i})}$$

After obtaining the percentages, they were again ordered from highest to lowest and the highest and lowest values were eliminated, so that possible errors in the experiment could be eliminated and the homogenization of the results of the samples would be more accurate. The remaining eight percentages were those included in the statistical software. The results are presented as average ± standard deviation.

### 4.5. Proliferation Test

Proliferation assays were performed using Trypan Blue staining agent (ddBiolab, cat. no. P08-34100). Cells were seeded on p60 cell culture plates at a final concentration of 200,000 cells/mL for all assays performed. They were exposed 24 h after seeding for 24 h. Once the exposure time had elapsed, cell–plate junctions were broken using trypsin enzyme (Thermo Fisher Scientific Inc., cat.no.15090046) by leaving it to act for 5 min at 37 °C. After incubation, the cells were removed and centrifuged at 1500 rpm for 5 min at 21 °C. The supernatant was removed and the cells were resuspended in 1 mL of medium. The LUNA II automatic cell counter (LogosBiosystems, Anyang, Republic of Korea) was used. The counter chamber was loaded with 12 µL of resuspended cells and Trypan Blue in a 1:1 ratio. The number of cells in each of the samples was ordered from smallest to largest in both populations (exposure/control) for each of the exposure conditions examined. The percentages of live and dead cell proliferation were calculated for each replicate (previously sorted) as follows:$${Proliferation\left(\%\right)}_{i}=\frac{Cell\;number({Exp}_{i})}{Cell\;number({Control}_{i})}\times 100$$

The data are represented as a value of increase or decrease in viability as follows:$${Proliferation(\%)}_{j}={Proliferation(\%)}_{i}-100\%$$

Three independent replicates were performed and 4 samples were taken per replicate, allowing a total of 12 proliferation data. All proliferation values are presented as average ± standard deviation.

### 4.6. Apoptosis Test

Apoptosis assays were performed using the Viability/Cytotoxicity Assay Kit for Animal Live & Dead Cells (Biotium, cat.no. 30002) which includes Calcein AM to stain live cells and EthD-III to stain apoptotic cells following the manufacturer’s specifications for fluorescence microscopy analysis. Cells were seeded at a final concentration of 100,000 cells/mL in the magnetic field strength homogeneity zone of a 96-well culture plate. Briefly, the culture medium in which the cells were immersed was removed. Then, they were stained with 2 µM calcein AM and 4 µM EthD-III in sufficient volume to cover the monolayer. They were allowed to stand for 30 min in the dark at room temperature. They were observed with a LEICA DFC340 FX fluorescence microscope (Danaher, Washington, DC, USA) and DFC Twain camera software (DFCTwain, v.6.9.0.107, Leica Microsystems, Heerbrugg, Switzerland). The calcein signal was visualized with the FITC filter sets, and the EthD-III signal was visualized with the TexasRed filter sets. The images were processed with FiJi software (ImageJ2, v.2.14.0, Madison, WI, USA). Eight replicates were carried out in two independent experiments. The percentage of live cells, the percentage of dead cells, and the total number of live cells were obtained. Values were ordered from lowest to highest, and the highest and lowest deviations were eliminated to homogenize the sample.

All experimental data are presented as average ± standard deviation.

### 4.7. Statistical Analysis

The results of the controls vs. samples exposed at 24, 48 and 72 h were analyzed with SPSS Statistics software (IBM SPSS Statistics© Software, v.29.0.0.0., New York, NY, USA). First, a Shapiro–Wilk normality test was applied to check the normality of the data with a 95% confidence interval. Data that passed the normality test (*p* > 0.05) were subjected to an F test to check the equality of variances or Levene’s test with a 95% confidence level. Tests that do not have a normal distribution were analyzed with a parametric Mann–Whitney U test with a 95% confidence level. Samples whose variance was similar (Levene’s test, *p* > 0.05) were analyzed with Student’s t-test for samples with similar variances with a 95% confidence level. If, on the other hand, the samples did not have an equal variance distribution (Levene’s test, *p* < 0.05), their statistical significance was analyzed using Student’s t-test for samples with unequal variances. Prism 9 software (GraphPad Software, v.9.3.1, Boston, MA, USA) was used for the graphical representation of the results.

## 5. Conclusions

It has been demonstrated that there are certain frequencies that can reduce the proliferation and viability of tumor cell models of nervous tissue. Exposure of cell models to different magnetic field frequency values produces a cellular response in viability and proliferation that responds to a “biological window” model centered at 50 Hz for the tumor models used. This window is cell-type specific. Astrocytes, the non-tumor cell line of comparison, show an increase in viability at 20 and 40 Hz. This allows us to consider the use of specific ’bioactive’ frequencies in therapeutic applications for different pathologies such as tumor development (with the consequent decrease in viability and proliferation) or neurodegenerative diseases (with the increase in viability and proliferation of the astrocytic network).

## Figures and Tables

**Figure 1 ijms-26-02032-f001:**
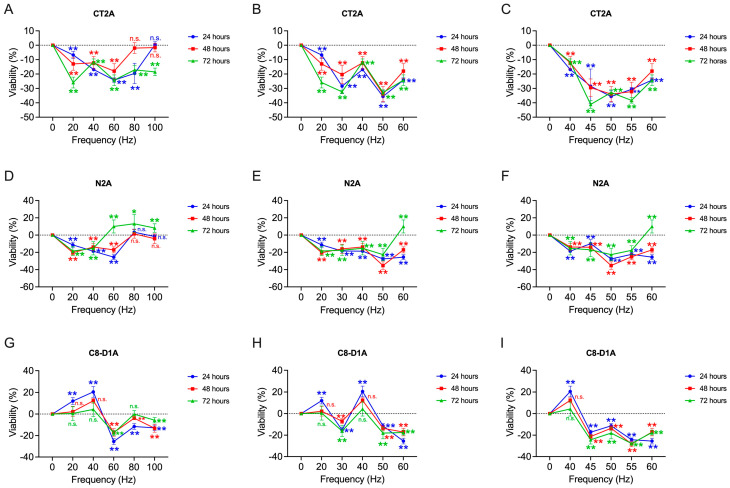
Percentage of viability obtained in the different frequency ranges tested, [20–100; 20] Hz (**A**,**D**,**G**), [20–60; 10] Hz (**B**,**E**,**H**) and [40–60; 5] Hz (**C**,**F**,**I**) of the different nerve tissue cell lines tested with respect to non-exposure controls. A, B, C: glioblastoma model (CT2A), (**D**–**F**): neuroblastoma model (N2A) and (**G**–**I**): astrocyte model (C8D1A). All assays are performed at a fixed intensity of 100 µT at 24, 48 and 72 h of exposure. Statistical results of application of the t-Student or U-Mann–Whitney statistical test with 95%-CI according to normality of the data (*) *p*-value < 0.05; (**) *p*-value < 0.001; (n.s.) non-significant.

**Figure 2 ijms-26-02032-f002:**
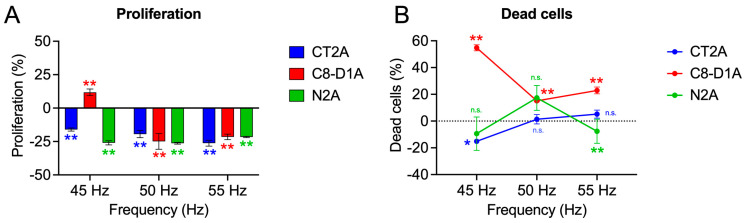
(**A**): Percentage of proliferation of the different nerve tissue cell lines tested with respect to non-exposure controls. (**B**): Percentage of dead cells with respect to non-exposure controls. It should be noted that 0 Hz represents the no-exposure controls. All tests are performed at frequencies of 45, 50 and 55 Hz with a fixed intensity of 100 µT in 24 h of exposure. Statistical results of the application of the Student’s t-test or Mann–Whitney U statistical test are presented with 95–CI according to the normality of the data. (*) *p*-value < 0.05; (**) *p*-value < 0.001; (n.s.) non-significant.

**Figure 3 ijms-26-02032-f003:**
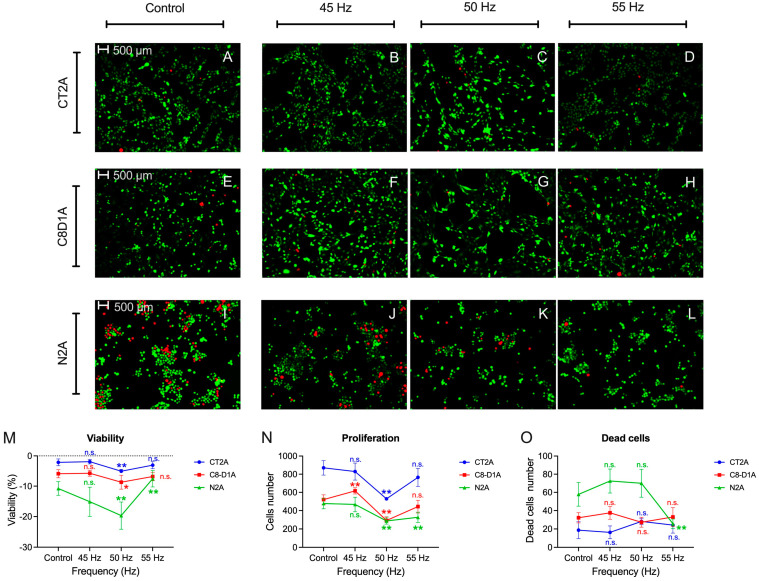
Apoptosis assay results at different frequencies of 45 Hz (**B**,**F**,**J**), 50 Hz (**C**,**G**,**K**) and 55 Hz (**D**,**H**,**L**) for CT2A (**A**–**D**), N2A (**E**–**H**) and C8D1A (**D**,**I**–**L**) tumor cell line models. The results of viability (**M**), proliferation (**N**) and dead cells (**O**) are presented. Statistical results of the application of Student’s t-test or Mann–Whitney U statistical test with 95–CI according to normality of the data: (*) *p*-value < 0.05; (**) *p*-value < 0.001; (n.s.) non-significant.

**Figure 4 ijms-26-02032-f004:**
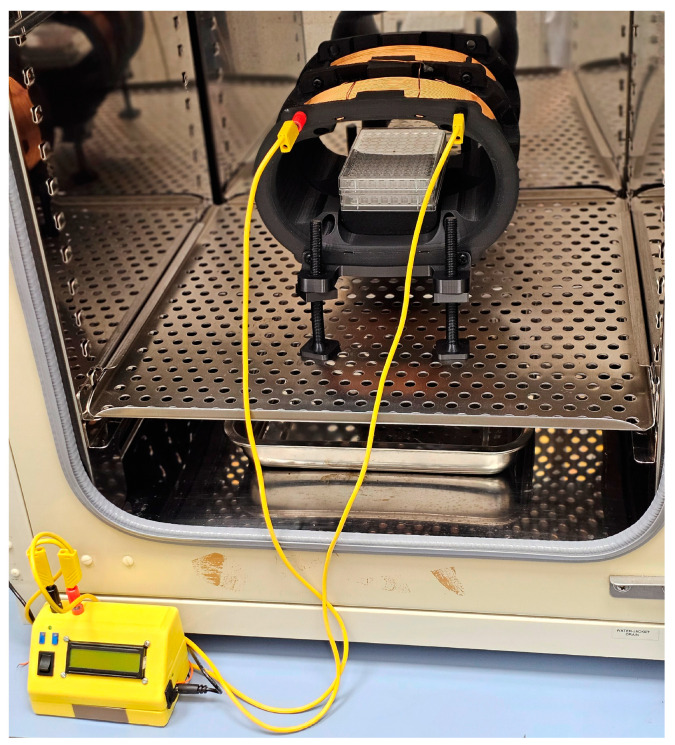
Image of the RILZ coil system used for cell experimentation.

**Table 1 ijms-26-02032-t001:** List of exposure parameters (intensity, frequency and time) and biological parameters (cell line, biomarkers) used in each of the experiments presented in this article.

Assay No.	Cell Line	Intensity [µT]	Frequency [Hz]	Time [Hours]	Biological Assay
1	CT2A, N2A, C8D1A	100	20, 40, 60, 80, 100	24, 48, 72	Metabolic activity (MTT)
2	CT2A, N2A, C8D1A	100	30, 50	24, 48, 72	Metabolic activity (MTT)
3	CT2A, N2A, C8D1A	100	45, 55	24, 48, 72	Metabolic activity (MTT)
4	CT2A, N2A, C8D1A	100	45, 50, 55	24	Proliferation and number of dead cells (Trypan Blue)
5	CT2A, N2A, C8D1A	100	45, 50, 55	24	Viability/Apoptosis (Calceína/EtDh)

## Data Availability

The raw data supporting the conclusions of this article will be made available by the authors on request.

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
