# Peer review of "The Frequency of a Magnetic Field Determines the Behavior of Tumor and Non-Tumor Nerve Cell Models"

_ijms, 2025, doi:10.3390/ijms26052032_

Round 1
Reviewer 1 Report
Comments and Suggestions for Authors
The study's results demonstrate that varying frequencies and time exposure altered cell viability in different cell lines. However, some points must be clarified.
1) What is the rationale for continuously exposing the cells for 24, 48, and 72 hours? The exposure time is exceptionally long if these data are associated with future therapeutic alternatives.
2) The title of the manuscript highlights the importance of frequency, but exposure time is also critical in the results. Besides, the effect according to the cell type is also different. Then, specifically, the frequency of a magnetic field determines the behavior of cells, could be questionable.
3) Figures 2 A and B should demonstrate the opposite results; less % of proliferation should show a higher cell death %, but the graphs seem contradictory, and this must be clarified. In C8D1A cells, at 45 Hz, the % of viability is low (Fig. 1I), and the % of cell death is high (Fig. 2 B), which is consistent; how do the authors explain the high % of proliferation (Fig. 2A) at the same stimulation conditions?
4) The method to determine the cell size must be detailed, and a protocol such as cell densitometry should be performed. The statement “No morphological alterations due to magnetic field exposure are shown” can’t be conclusive with this method; the analysis of the cytoskeleton should be performed.
5) The authors could represent the results using heatmaps or 3D surface plots to provide a clearer visual of the combined effects of frequency, time, and cell type on viability. To improve clarity for readers, graphs could be marked with critical insights, such as frequencies with the most statistically significant effects.
6) While the discussion points out multiple references concerning the cell molecular mechanisms affected by magnetic field exposure, the authors need to clarify why slight variations in frequency and exposure duration yield different effects on the cells.
Reviewer 2 Report
Comments and Suggestions for Authors
1) A sketch or photograph of the experimental situation would be very helpful,
2) Write more clearly (explicitly) what constitutes a control group in each situation, to which you then statistically evaluate the degree of significant difference.
3) Review the comments in the pdf file.

The language should be edited.
